# Empirical Quantification of Optic Nerve Strain Due to Horizontal Duction

**DOI:** 10.3390/bioengineering10080931

**Published:** 2023-08-05

**Authors:** Seongjin Lim, Joseph L. Demer

**Affiliations:** 1Department of Mechanical Engineering, University of California, Los Angeles, CA 90095, USA; seongjinlim@mednet.ucla.edu; 2Ophthalmology Department, University of California, Los Angeles, CA 90095, USA; 3Stein Eye Institute, Bioengineering Department, Neurology Department, Neuroscience Interdepartmental Program, University of California, Los Angeles, CA 90095, USA

**Keywords:** biomechanics, eye movement, magnetic resonance imaging, optic nerve

## Abstract

Magnetic resonance imaging (MRI) was used to measure in vivo local strains in the optic nerve (ON) associated with horizontal duction in humans. Axial and coronal MRI were collected in target-controlled gazes in 24 eyes of 12 normal adults (six males and six females, 59 ± 16 years) during large (~28°) and moderate (~24°) ductions. The ON, globe, and extraocular muscles were manually identified, and the pixels were converted to point-sets that were registered across different imaging planes and eye positions. Shape of the ON was parameterized based on point-sets. Displacements and strains were computed by comparing deformed with initial ON configurations. Displacements were the largest in the most anterior region. However, strains from adduction were uniform along the length of the ON, while those during abduction increased with distance from the globe and were maximal near the orbital apex. For large gaze angles, ON strain during abduction was primarily due to bending near the orbital apex that is less transmitted to the eye, but during adduction the ON undergoes uniform stretching that transmits much greater loading to the posterior eye, implied by greater strain on the ON.

## 1. Introduction

Biomechanics of the optic nerve (ON) have elicited interest because mechanical stimuli acting on the eye may contribute to development of ocular diseases [1,2,3]. Hydrostatic pressure within the eye has been primarily investigated as an influential factor [3]. Because large adduction tethers the ON and is associated with mechanical deformations [4] of the optic disc and peripapillary region, recent studies have investigated biomechanical effects of ocular rotation [4,5,6,7,8] using finite element analysis (FEA) [7,8,9,10,11], optical coherence tomography (OCT) [6,12], and magnetic resonance imaging (MRI) [5,13]. However, FEA necessarily requires simplifications, so resulting simulations only predict accurate mechanical responses to the extent that all aspects of ocular kinematics and tissue properties are accurately implemented. In particular, most prior FEA studies have simulated horizontal duction without considering globe translation [7,9], although it is has recently become clear that the eye translates as it rotates [14].

Except for theoretical FEA, however, local strains in the ON have not previously been explored. Some strains inside the eye can be measured by OCT, but its penetration depth is limited to only a few hundred microns [15], which is insufficient to capture the retrobulbar ON. On the other hand, MRI can visualize deep posterior regions of the eye and orbit, albeit with lower resolution. Several MRI studies have investigated stretching of the ON by horizontal eye rotation, typically reporting variations in tortuosity [5,13,14] or length [16]. For other organs, MRI has also been used to measure displacements and strains, but low resolution has precluded application to the ON.

We circumvented previously encountered MRI resolution limitations using image combination and parameterization of ON geometry that permits in vivo analysis of ON strains caused by eye rotation. Because of the long imaging time required, the accumulation of motion artifacts produced by involuntary eye movements during eccentric gaze typically preclude isotropic orbital MRI at high spatial resolution. It has long been our approach to obtain high resolution of about 312 µm in single orientation planes using surface coils, but plane thickness is typically 2 mm [17]. However, sequentially imaging with a rest period between acquisitions in two orthogonal image planes under identical viewing conditions minimizes motion artifacts for each acquisition. We used post-processing to combine high-resolution acquisitions in orthogonal planes to define 3D point clouds delineating the ON. Then, we parameterized a curve defining the ON centroids using an iterative algorithm [18] enabling reconstruction of continuous 3D ON structure. By applying our method to ocular images from healthy people, we quantified the mechanical deformations of the ON using an automated process. We used this approach to develop a theoretical framework to understand mechanical loading on the ON during eye rotation.

## 2. Materials and Methods

### 2.1. Human Subjects and MRI Acquisition

This research was approved by the Institutional Review Board for Protection of Human Subjects of the University of California, Los Angeles, and conforming to the tenets of the Declaration of Helsinki, subjects gave written informed consent before participation.

For MRI images, 24 eyes of 12 healthy adults (6 male and 6 female, mean 59 ± 16 years) were imaged using Signa 1.5T scanner (GE Healthcare, Chicago, IL, USA) equipped with ocular surface coils (Medical Advances, Milwaukee, WI, USA), and employing T2 fast spin-echo weighting [17]. Gaze was controlled by target fixation by the centered or abducting eye using the detailed protocols and image resolution we have published [5]. The target location was centered for central gaze images, and moved laterally by different amounts for moderate or large abducted or adducted gazes. Central gaze refers to the eye position fixating a target directly in front of the eye. The intended (and achieved average) increment between moderate and large duction was about 6°, but this was difficult to determine with precision in individual cases because of individual subject differences in distance from the eye to fiberoptic target on the surface of the surface coil mask. For adduction, fixation was by the fellow eye. As a result, MRI data for central gaze, large and moderate adduction, and large and moderate abduction were obtained (Figure 1). Imaging was performed in axial planes, as well as quasi-coronal planes perpendicular to the long axes of each orbit separately. Images from some subjects have been previously analyzed for other purposes and reported in other studies [16,19].

### 2.2. 3D Reconstruction and Point-Set Registration

For MRI segmentation, we manually identified the globe, ON, orbital wall, lateral rectus (LR), and medial rectus muscles (MR) in quasicoronal and axial images (Figure 2A). Then, the images were interpolated by video frame interpolation [20] along the long orbital axis. The segmented pixels from the image stack resulting from each acquisition were converted to 3D point-sets corresponding to the gaze direction during imaging.

Rigid point-set registration [21] was performed between coronal and axial image stacks (Figure 2B). Because the quasicoronal image stacks spanned the length of the ON but did not completely span the anterior globe, we used point clouds that correspond to the completely-imaged regions of the ON, LR, and MR for this registration. Point clouds from LR and MR muscles were used to resolve any ambiguity due to axisymmetric ON geometry. Finally, we registered point clouds between central gaze and horizontal duction, reasonably assuming that points for the rigid bony orbital wall are not displaced by eye rotation.

### 2.3. Curve Parametrization

After point-set registration, we combined point clouds from axial and quasicoronal MRI to identify ON centroids along its entire orbital length. Defining a convex hull from points on the globe surface, the intersection between the convex hull and a line through the most anterior two ON centroids was regarded as the ON junction (Figure 2C). The normal vector to the hyperplane that includes the ON junction defined the gradient of ON trajectory at the ON junction. Because the orbital apex was not definable in MRI, we found it necessary to operationally define its pseudo-location in the following manner. During large adduction, the ON is stretched straight, so a line defining ON path would be directed a fixed origin at the mouth of the optic canal in the orbital apex. By extrapolating the orbital wall, we identified the point where the straightened ON intersects the projection of the orbital wall, and considered this point to be the orbital apex. Although this point is anatomically artificial, it is fixed near the orbital apex, and thus useful for measuring changes in remainder of the ON caused by eye rotation. The ON centroids, the ON junction with the globe, and the pseudo-orbital apex became knot points for the parameterization. Optic nerve path defined by parametric curve fitting to the knot points was performed by minimizing distances between the curve and the points [18].

### 2.4. Local Coordinate System for the Globe and Globe Translation

Mutual co-registration allowed bilateral registration of quasicoronal MRI for the left and right eyes, to axial images (Figure 3). We identified the medial plane of the head by minimizing the Chamfer distance between the reflected point-sets between the two eyes. Chamfer distance is a metric to compute the similarity between two point clouds [22], so we could estimate the best-fitted medial plane whose normal vector was defined to be the *X*-axis. From coronal MRI, we identified in sequential quasi-coronal images the junction of the superior ethmoid air sinus and orbit as the anterior ethmoid recess [23]. Then, we rotated the line connecting the junctions in each orbit by 10° in the medial plane to define the *Z*-axis. The *Y*-axis was determined by the cross-product between the *X*- and *Z*-axes (Figure 3). By using the resulting standardized coordinate system-based anatomy, we could compare data among different subjects and orbits.

The minimal bounding sphere for the eye was defined from quasicoronal images, and the center of the sphere was considered the globe center. Globe translation was obtained by the comparison of its center points before and after duction.

### 2.5. Duction Angle Measurements

From the point cloud representing the ocular surface, we identified the corneal apex and the antipodal point most posterior to the apex. Ocular orientation was operationally defined by the line spanning from the corneal apex to the antipodal point, while the positive angle indicates abduction. The angle between ocular orientation and the globe’s *Z*-axis was considered the duction angle. The defined eye orientation neglects the individually variable visual angle kappa, but accurately captures changes in eye orientation from initial position. We measured the angle of ocular torsion (twisting along the line of sight) from positions of the LR and MR muscle insertions on the globe.

The subjective visual direction (line of sight) is not equivalent to the line of anatomical symmetry of the eyeball. The angular difference between these two lines is termed the “angle kappa” [24,25]. Angle kappa is individually variable but usually positive, meaning that an eye may appear anatomically abducted when in fact it is looking straight ahead. Anatomical angles reported here are offset by angle kappa.

### 2.6. Local Coordinate Systems for the ON and the Reconstruction of the ON

We defined the ON junction with the globe as the origin of a local coordinate system whose *X*-axis is defined by the line between the antipode of the corneal apex, and ON junction. The *Z*-axis (posterior-anterior) parallels the normal vector of the globe surface at the ON junction, and the *Y*-axis is orthogonal to the *X*- and *Z*-axes. This enabled standardization of data using a coordinate system based on biological landmarks. In addition, we defined a local coordinate system at each ON centroid along its path. The local *Z*-axis is parallel to the tangent vector of the parameterized curve, and the local *X*-axis is parallel to the projection of the local *X*-axis defined at the ON junction on the plane whose normal vector is the local *Z*-axis. The local *Y*-axis is perpendicular to both the local *X*- and *Z*-axes. Based on the local coordinates system and the parameterized curves defining ON paths, we could identify corresponding matched points for the ON for different ductions (Figure 2B).

Because the ON path was a parameterized curve, its tangent vector was readily computed. Then, we defined a set of planes whose normal vectors are tangent to ON path and originate at the corresponding points in the ON path. By the projection of the adjacent ON outlines onto this plane, ON cross-section in each plane was determined, allowing computation of the convex hull for the ON border. We could then parameterize the local polar coordinates in the plane of the convex hull defining the border of the ON cross-section. The volume of ON was measured by a summation of the products of cross-sectional area and the length of line segment that divides the ON path with equal length.

### 2.7. Displacements and Strains

Corresponding points in the initial and deformed configurations were identified from the ON path parameterization. Since below we found insignificant torsion change implying invariance of the angular coordinate for each point, the radial coordinate could be normalized by the largest radius along each radial line. The local polar coordinates for the ON centroid supported the idea that points on different configurations could be compared. By subtracting the deformed from initial coordinates, we obtained displacements. Gradients of the displacements are required for computing the Green–Lagrange strain, which represents $E=(\nabla u^{T}+\nabla u+\nabla u^{T}\cdot \nabla u)$ where ∇u and $\nabla u^{T}$ are the displacement gradient and its transpose [26]. However, because point clouds represent unstructured data, gradients along the *X*-, *Y*-, and *Z*-axes could not be obtained directly. Instead, we found 6 nearest neighbors of each point and computed the directional derivatives along vectors from the point to 6 nearest points (Figure 2D). Since the *X*-, *Y*-, and *Z*-axes can be represented by linear combination of these 6 vectors, we obtained the gradient of the displacements from the linear combination of the directional derivatives. The unit vector parallel to each axis can be represented by linear combination of 6 nonparallel vectors $e=k_{1}v_{1}+k_{2}v_{2}+k_{3}v_{3}+k_{4}v_{4}+k_{5}v_{5}+k_{6}v_{6}$. The equation can be rewritten in a matrix form as follows. *e* = VK where e is the unit vector for each axis, V is a matrix whose column vectors are normalized vectors between each point and 6 nearest points, and K is a matrix whose elements are undetermined coefficients for the linear combination. Then, V^T^e = V^T^VK where V^T^ is the transpose of matrix V. As a result, the coefficients K could be determined using the pseudo inverse matrix. K = (V^T^V)^+^V^T^e where (V^T^V)^+^ is a pseudo inverse matrix of V^T^V. The computed displacements and strains were grouped into 5 regions of initially equal length along the ON (Figure 2D), with region G1 closest to the globe-ON junction, and G5 closest to the orbital apex.

### 2.8. Measurements of ON and Optic Nerve Sheath (ONS) Diameters

For quasicoronal images, best-fitted circles to include the ON, cerebrospinal fluid, and ONS were drawn manually by using ImageJ [27] (Wayne Rasband, National Institutes of Health, Bethesda, MD, USA). Diameters were computed from cross-sectional areas of the circles, and the thickness of the enclosed cerebrospinal fluid gap was used to obtain the inner diameter of the ONS. The inner and outer diameters of ONS were normalized by dividing them by the ON diameter, and the ratio in central gaze was compared to that in large adduction for the same eye.

### 2.9. Statistical Analysis

Normality testing and Mann–Whitney testing were conducted using GraphPad Prism 9 (GraphPad Software, San Diego, CA, USA). Generalized estimating equation (GEE) analysis using IBM SPSS Statistics 25 (IBM, Armonk, NY, USA); this approach accounts for possible correlations between the two eyes of the same subjects and has type 1 error characteristics better than *t*-testing [28].

## 3. Results

### 3.1. Changes in Geometry Due to Eye Rotation

Mean (±standard error, SE) horizontal eye orientation where the positive sign represents abduction, was 9.8 ± 1.1° in central gaze, −28.1 ± 1.0° in large adduction, −23.8 ± 0.9° in moderate adduction, 30.4 ± 1.3° in large abduction, and 24.2 ± 1.0°, in moderate abduction, respectively (Figure 4A). Tortuosity is defined to be the ratio of actual ON path length divided by the straight line (minimal) distance between the ON junction and orbital apex. Tortuosity for central gaze was 1.023 ± 0.004, for moderate adduction 1.017 ± 0.002, for large adduction 1.016 ± 0.002, for moderate abduction 1.029 ± 0.003, and for large abduction 1.036 ± 0.004 (Figure 4B). These results indicate that adduction straightens the ON path, while abduction does not.

Ocular torsion angle changed by 1.6 ± 0.6° for moderate adduction, 0.6 ± 0.6° for large adduction, 0.3 ± 0.4° for moderate abduction, and 0.1 ± 0.6° for large abduction (Figure 4C). Because none of these measured torsion angle changes differed significantly from zero, we considered ON twisting negligible during horizontal duction.

### 3.2. Globe Translation

We measured the translation of the globe centroid after duction. Data are plotted for horizontal translation in Figure 5A. For large adduction, the globe translated medially by 0.69 ± 0.09 mm. For large abduction, the globe laterally translated 0.58 ± 0.08 mm. For moderate adduction, the globe translated nasally by 0.58 ± 0.09 mm. For moderate abduction, the globe translated temporally by 0.45 ± 0.06 mm. All of these horizontal globe translations differed significantly from zero (*p* < 0.0001, Mann–Whitney). Corresponding small superior (Y axis) translations were −0.08 ± 0.08 mm, 0.10 ± 0.10 mm, −0.09 ± 0.09 mm, and 0.09 ± 0.07 mm, respectively; none of these differed significantly from zero (Figure 5B).

There was modest posterior globe translation for abduction (Figure 5C). Posterior globe translation was 0.29 ± 0.14 mm during moderate adduction, 0.25 ± 0.12 mm during large adduction, −0.23 ± 0.11 mm during moderate abduction, and 0.53 ± 0.10 mm during large abduction. Translation did not differ significantly from zero for adduction, but was significant for large and moderate abduction (*p* < 0.005, GEE).

In addition, we measured the magnitude of combined 3D globe translation during ductions (Figure 5D). The 3D translation magnitudes were 0.79 ± 0.07 mm for moderate adduction, 1.03 ± 0.09 mm for large adduction, 1.01 ± 0.11 mm for moderate abduction, and 1.04 ± 0.09 mm for large abduction. Large ductions resulted in larger magnitudes of translation than moderate ductions (*p* < 0.05, GEE).

### 3.3. Local ON Displacements

We measured local 3D displacements of the ON due to duction in five regions of equal length along its anteroposterior extent. As seen in Figure 6, displacements were greatest anteriorly and progressively decreased posteriorly; this effect was significant for large and moderate ab- and adduction (*p* < 0.001). Moreover, adduction induced larger displacements than abduction (*p* < 0.05, GEE).

### 3.4. Local ON Strain Due to Duction

We computed ON strains by differentiating local ON displacements for the direction tangent to ON path in the five ON regions defined above (Figure 7). Local strain behavior differed fundamentally between ad- and abduction. Strains were about 5% for large and 4% for moderate adduction, but did not vary significantly by region for either direction (*p* > 0.9, GEE). In contrast, for both moderate and large abduction, strain was least at about 1% in region G1 that abutted the globe, and increased monotonically to about 4% in region G5 at the orbital apex. For abduction, strain did not depend significantly on duction size, but did depend on region (*p* < 0.05, GEE). For large adduction, strains were about 0.055 in all regions (Figure 7A), while in large abduction, strains increased monotonically from 0.011 ± 0.013, in region G1 to 0.037 ± 0.010 in region G5 (Figure 7B). Moderate adduction resulted in ON strain of about 0.04 in all regions (Figure 7C), while in moderate abduction, strain values increased monotonically from 0.014 ± 0.011 in Region G1 to 0.046 ± 0.011 in Region G5 (Figure 7D).

### 3.5. Optic Nerve and Sheath Diameters during Large Adduction

The ratios of inner and outer diameters of the ONS relative to ON diameter were not affected by adduction (Figure 8; *p* > 0.6 for each, GEE). The relative ONS inner and outer diameters in central gaze were 1.55 ± 0.03 and 1.91 ± 0.04. Corresponding values were 1.56 ± 0.02 and 1.86 ± 0.04 in large adduction.

## 4. Discussion

This study combined high-resolution quasicoronal and axial MRI to determine precisely the geometric changes in the globe and ON associated with horizontal duction in normal adult humans. The current findings in Figure 5A confirm and extend the finding of Clark and Demer that the globe translates nasally in adduction and temporally in abduction [14], in the present case finding nasal translation in adduction to significantly exceed temporal translation in abduction. The current analysis also demonstrates significant posterior globe translation averaging up to 0.5 mm in abduction, but none in adduction (Figure 5C). This finding differs modestly from the report of Clark and Demer, who analyzed quasicoronal MRI only and reported no posterior translation in either add- or abduction in normal adults [14]. This difference may be because Clark and Demer computed globe translation without whole orbital registration between eye positions, whereas the current study registered images by referring the entire bony orbit. Since the rectus muscle pulleys that serve as the functional mechanical origins are close to globe center, globe translations during rotation have significant kinematic implications for required forces and neural control strategies during eye movements [19,29].

The current study also confirms the prediction of Wang et al. that abduction would also produce ON strain despite its sinuous path in this gaze direction [7]. It is intuitively simpler to understand ON elongation strain in large adduction where the ON path remains straight. Sinuosity of the ON path presents a more complex situation. We interpret strain in the sinuous path of the ON in abduction from the perspective of solid mechanics, finding it analogous to the bending of a cantilever under a load at the free end. In this situation, the fixed end of the cantilever undergoes greatest strain, while least strain occurs at the free end. Anatomically, the posterior ON is fixed near the orbital apex, while the anterior end is translated by the rotating globe. Correspondingly, the current findings indicate that sinuous ON strain increases with proximity to the orbital apex, and is minimum at the ONH (Figure 7). This behavior suggests that abduction would perturb the optic disc less than adduction, and is consistent with the proposed special role of repetitive adduction tethering of the ON in the pathology of primary open angle glaucoma [5,16]. Because of the asymmetric location of the ON junction, adduction and abduction behave differently. As a result, adduction straightens the ON path, and abduction increases the tortuosity of the ON path (Figure 4B).

This study demonstrated using high-resolution MRI the effect of horizontal eye rotation on strains in the ON. The resulting strains may be understood as beam bending in abduction, but tensile elongation in large adduction. The findings suggest implications of mechanical deformations of the eyeball itself caused by ON strain, particularly during adduction. The image analysis techniques described here could be applied to subjects with glaucoma and strabismus to evaluate the suggested role of ON strain in pathogenesis of optic neuropathy.

## Figures and Tables

**Figure 1 bioengineering-10-00931-f001:**
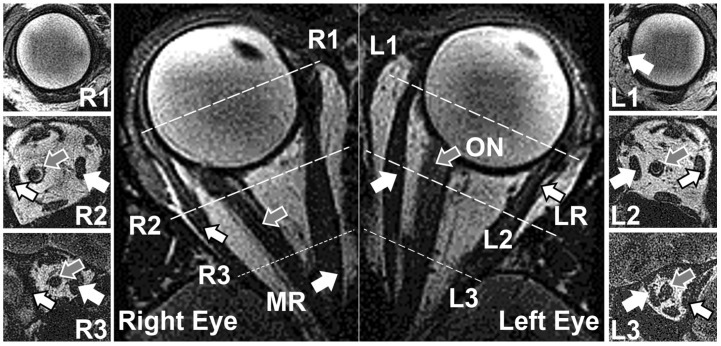
Representative quasicoronal (resolution 312 µm) and axial (resolution 390 µm) MRI for a subject in large angle left gaze. R1, R2, and R3 images represent anterior, medial, and posterior section of the right orbit, obtained in the planes indicated by the dotted lines. Similarly, L1, L2, and L3 correspond to anterior, medial, and posterior section of the left orbit. MR—medial rectus muscle. LR—lateral rectus muscle. ON—optic nerve. White arrors indicate MR, and arrows with black edges represent LR. Gray arrows point to ON.

**Figure 2 bioengineering-10-00931-f002:**
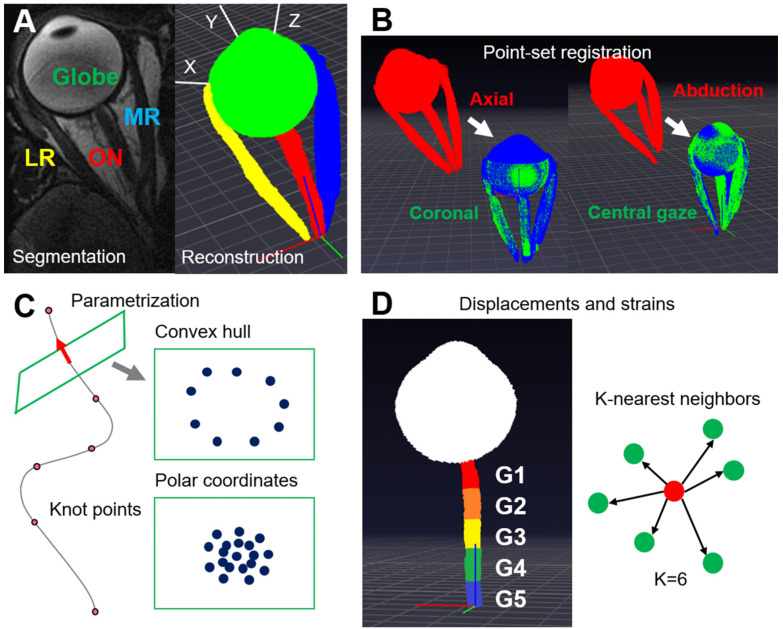
Workflow to measure displacements and strains due to horizontal duction. (**A**) Globe, ON, and extraocular muscles are segmented from MRI, and pixels in the image stack are converted to 3D point-sets. The defined coordinate system is drawn. (**B**) Point-set registration between axial and quasicoronal images, and between different eye positions. Red point-sets are registered to green point-sets, and blue point-sets are results of the registration. White arrows represent registered point-sets. (**C**) Centroids of ON segments serve as knot points for a curve to parameterize ON path, and a local polar coordinate system on the plane normal to the curve parameterizes ON cross-section. (**D**) Displacements of the ON are computed by comparing deformed and initial configurations. The gradients of displacements calculated by comparing 6 nearest neighbor points determine strains. Data are divided into 5 equal segments along ON length.

**Figure 3 bioengineering-10-00931-f003:**
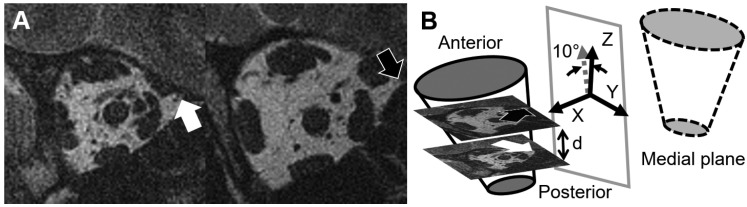
Definition of the coordinate system. (**A**) Quasicoronal MRI from a right eye. The arrows represent the anterior ethmoid recess in two different image planes. (**B**) By reflecting the orbital wall point-set, a medial plane is defined, and its normal defines the *X*-axis. Because coordinates of the anterior ethmoid recess and the distance between image planes are known, the pitch angle of the orbit can be calculated. The gray dashed arrow represents the vector between 3D coordinates of the anterior ethmoid recess in separate quasi-coronal image planes, and the vector‘s direction is used to define *Z*-axis after rotating 10° in the medial plane. The *Y*-axis is orthogonal to the *X*- and *Z*-axes, and is positive towards the top of the head.

**Figure 4 bioengineering-10-00931-f004:**
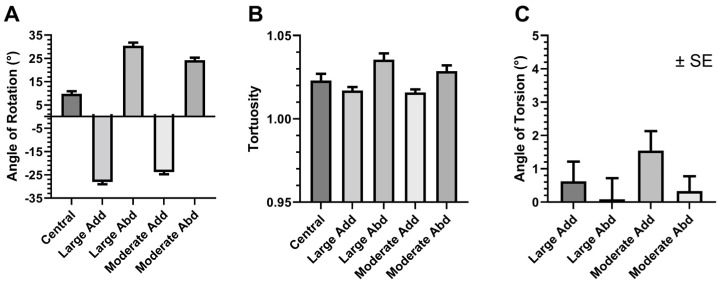
Changes due to horizontal duction. (**A**) Eye orientation relative to the medial plane of the head. (**B**) Tortuosity, the ratio of actual ON length to length of straight line from the ON junction to the orbital apex. (**C**) Ocular torsion did not change significantly with duction. SE—standard error of the mean. Abd—abduction. Add—adduction.

**Figure 5 bioengineering-10-00931-f005:**
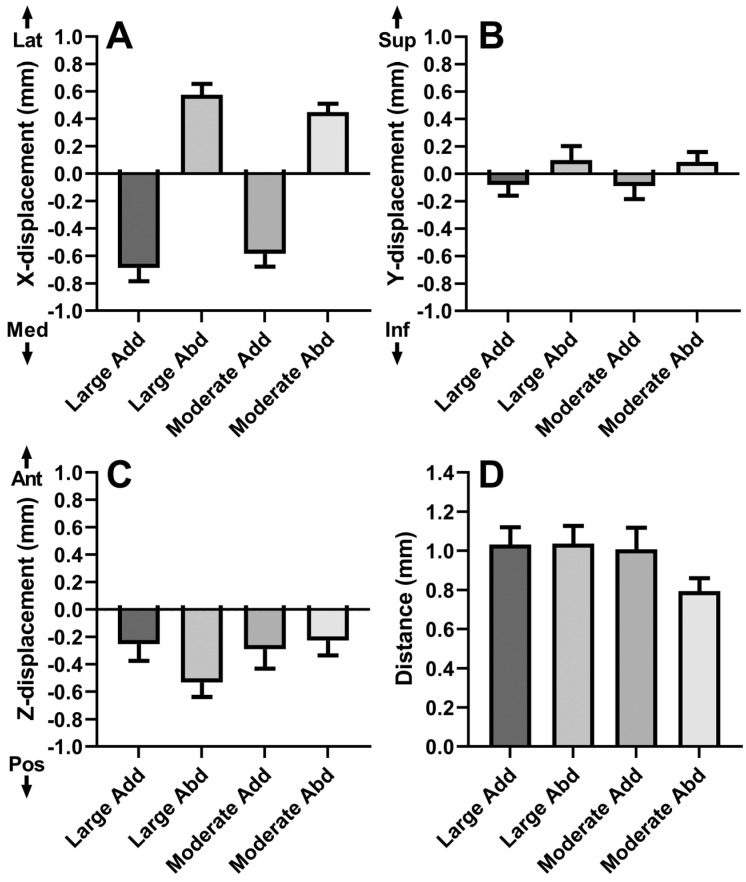
Globe translation after horizontal duction. (**A**) Mediolateral. (**B**) Vertical (**C**) Anteroposterior. (**D**) Magnitude of 3D translation. Abd—abduction. Add—adduction. SE—standard error of the mean.

**Figure 6 bioengineering-10-00931-f006:**
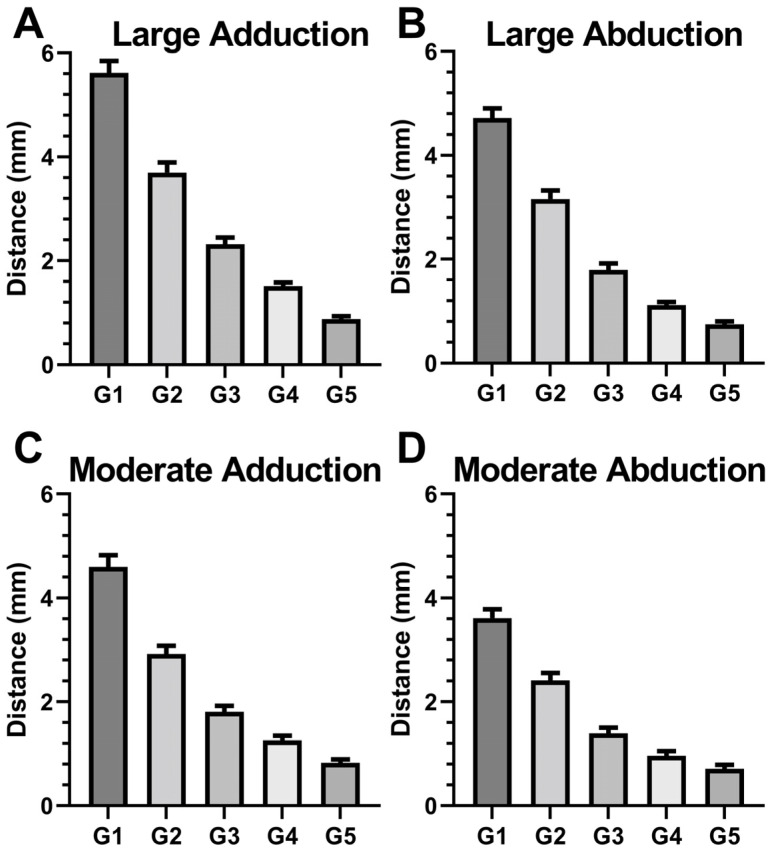
Regional 3D displacements of the optic nerve (ON) for (**A**) large adduction, (**B**) large abduction, (**C**) moderate adduction, and (**D**) moderate abduction. G1 through G5 are equal intervals along the ON length from anterior to posterior, respectively. SE—standard error of the mean.

**Figure 7 bioengineering-10-00931-f007:**
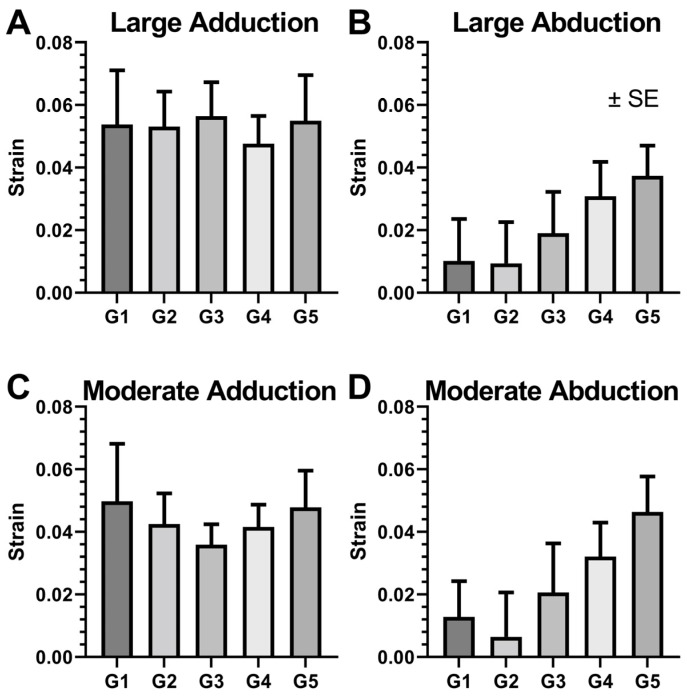
Local strains along the path of the optic nerve (ON) depending on the ON region for (**A**) large adduction, (**B**) large abduction, (**C**) moderate adduction, and (**D**) moderate abduction. Strain did not vary significantly along the length of the ON in adduction, but increased progressively towards the orbital apex for both large and moderate abduction. G1 through G5 are equal intervals along the ON length from anterior to posterior, respectively. SE—standard error of the mean.

**Figure 8 bioengineering-10-00931-f008:**
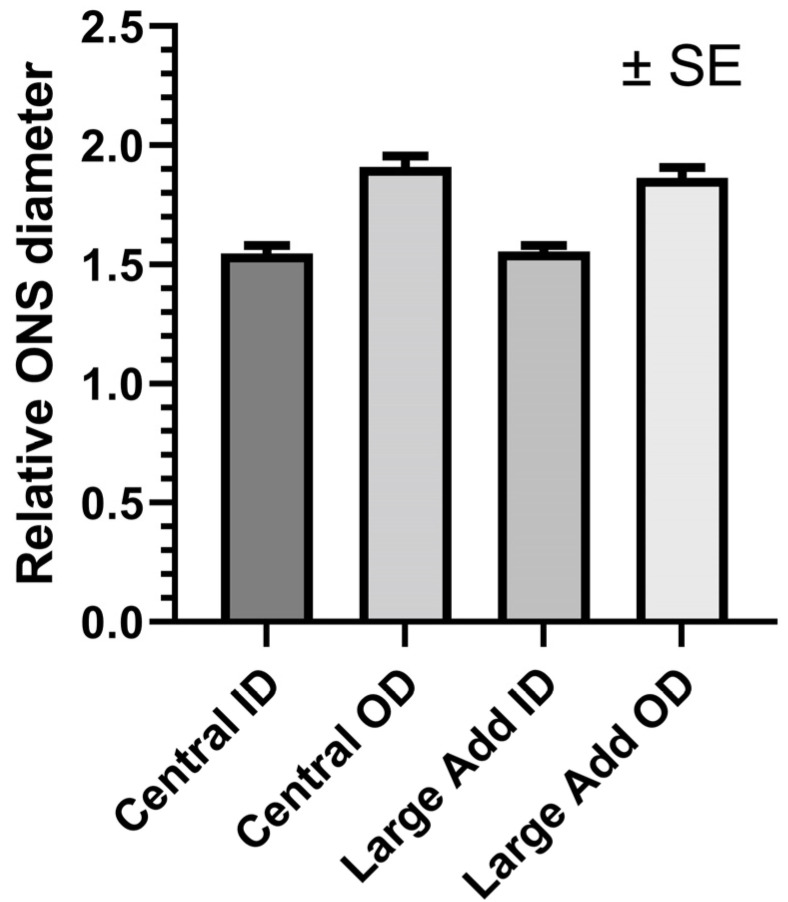
Duction did not significantly alter relative optic nerve sheath (ONS) diameters. ID—inner diameter. OD—outer diameter. *p* > 0.6. SE—standard error of the mean.

## Data Availability

Not applicable.

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
