# Peer review of "Empirical Quantification of Optic Nerve Strain Due to Horizontal Duction"

_bioengineering, 2023, doi:10.3390/bioengineering10080931_

Round 1

Reviewer 1 Report

Lin and Demer inferred from MRI imaging the strain on the optic nerve when subjects are holding eccentric eye orientations. While not ground breaking per se, the results add to our growing understanding of how mundane activities such as looking out to the side might play a role in the genesis of glaucoma.

I found the manuscript to be well written and easy to understand, with a few minor exceptions listed below.

Ln. 13             I found the “small” vs “large” labeling for 24 vs 28 deg ductions peculiar. How about using “moderate” vs “large” instead, or something to that effect? That would apply throughout, of course.

Ln. 71             Centered relative to what? I think that there is a need to expand the methods here, as the reader should be able to understand where exactly the target is relative to each eye without having to consult another publication.

Fig. 1              I found this figure somewhat confusing, as it is not clear what one should look at. The axial images give a good idea of the quality of the imaging, but it’s hard to understand what one should make of the coronal slices. Perhaps some arrows could be added to indicate features that are important to the study’s conclusions?

Ln. 87             Presumably this step was done on coronal slices?

Ln. 112           cause should be caused

Ln. 117           This paragraph is not very clear. It would be helpful to have the axes shown either on a drawing or on an MRI image, so that it is clear in which direction they point. They could even be superimposed in Fig. 1.

Ln. 147           form?

Ln. 150           Same as for Ln. 117, it would be helpful to indicate the axes in a drawing (or in Fig. 2)

Ln. 185           unparalleled?

Ln. 212           9.8 deg of abduction are reported in central gaze. I presume that this is the non-viewing eye, although if this is so it would need to be spelled out (and it would be confusing given that at line 77 we are given the impression that only adductions are from the non-viewing eye). The reader should not have to guess which measurements come from the viewing eye.

Fig. 3              Eye rotations in the text are signed, but here they are all positive. Why? It’s confusing.

Ln. 297           Just as 24 deg is not what most people would consider a small eye rotation, 59 years of age is not what most people would consider young. I know it’s all relative, but still…

Ln. 311           produces should be produce

Ln. 320           load perturb?

Ln. 323           behaves should be behave

Ln. 331           There’s an extra “that”

Author Response

Reply to Reviewer 1

I found the manuscript to be well written and easy to understand, with a few minor exceptions listed below.

REPLY: Thank you.

Ln. 13             I found the “small” vs “large” labeling for 24 vs 28 deg ductions peculiar. How about using “moderate” vs “large” instead, or something to that effect? That would apply throughout, of course.

REPLY: Done throughout.

Ln. 71             Centered relative to what? I think that there is a need to expand the methods here, as the reader should be able to understand where exactly the target is relative to each eye without having to consult another publication.

REPLY: A detailed explanation has been added as requested, at lines 74 and 75 of the revision.

Fig. 1              I found this figure somewhat confusing, as it is not clear what one should look at. The axial images give a good idea of the quality of the imaging, but it’s hard to understand what one should make of the coronal slices. Perhaps some arrows could be added to indicate features that are important to the study’s conclusions?

REPLY: The figure has been revised and extensively labeled for clarity.

Ln. 87             Presumably this step was done on coronal slices?

REPLY: Line 93 of the revision now indicates that segmentation was performed on both quasicoronal and axial images.

Ln. 112           cause should be caused

REPLY: Typo has been corrected in line 118.

Ln. 117           This paragraph is not very clear. It would be helpful to have the axes shown either on a drawing or on an MRI image, so that it is clear in which direction they point. They could even be superimposed in Fig. 1.

REPLY: For adequate clarity, we have added new Fig. 3 and additional description in its legend. Lines 149 – 159 of the revision.

Ln. 147           form?

REPLY: Typo has been corrected at lines 167-168.

Ln. 150           Same as for Ln. 117, it would be helpful to indicate the axes in a drawing (or in Fig. 2)

REPLY: The figure has been revised to represent the axes.

Ln. 185           unparalleled?

REPLY: Revised to “nonparallel.” (line 210)

Ln. 212           9.8 deg of abduction are reported in central gaze. I presume that this is the non-viewing eye, although if this is so it would need to be spelled out (and it would be confusing given that at line 77 we are given the impression that only adductions are from the non-viewing eye). The reader should not have to guess which measurements come from the viewing eye.

REPLY: This is actually the position of the viewing eye with the target centered in front of it, and is the result of the “angle kappa” phenomenon whereby the subjective line of sight is modestly nasal to apparent anatomical eye orientation. The sentences below are added in the manuscript for the clarity.

The subjective visual direction (line of sight) is not equivalent to the line of anatomical symmetry of the eyeball. The angular difference between these two lines is termed the “angle kappa” (Basmak et al. 2007; Gharaee et al. 2014). Angle kappa is individually variable but usually positive, meaning that an eye may appear anatomically abducted when in fact looking straight ahead.

Fig. 3              Eye rotations in the text are signed, but here they are all positive. Why? It’s confusing.

REPLY: We take your point, and have used opposite sign for opposite rotations. The figure has been re-numbered to Fig. 4 in the revision.

Ln. 297           Just as 24 deg is not what most people would consider a small eye rotation, 59 years of age is not what most people would consider young. I know it’s all relative, but still…

REPLY: We have deleted the word “young” in line 321 of the revision. We just do not feel all that old…

Ln. 311           produces should be produce

REPLY: Typo has been corrected in line 335.

Ln. 320           load perturb?

REPLY: The redundant “load” has been deleted at line 344 of the revision.

Ln. 323           behaves should be behave

REPLY: Typo has been corrected at line 347 of the revision.

Ln. 331           There’s an extra “that”

REPLY: The unnecessary word has been deleted at line 355.

Reviewer 2 Report

The paper's structure is well organized, and the authors reasonably designed the experiment.

Author Response

Reply to Reviewer 2

The paper's structure is well organized, and the authors reasonably designed the experiment.

REPLY: Thank you.